# Dietary Supplements for Male Infertility: A Critical Evaluation of Their Composition

**DOI:** 10.3390/nu12051472

**Published:** 2020-05-19

**Authors:** Andrea Garolla, Gabriel Cosmin Petre, Francesco Francini-Pesenti, Luca De Toni, Amerigo Vitagliano, Andrea Di Nisio, Carlo Foresta

**Affiliations:** 1Unit of Andrology and Reproductive Medicine & Centre for Male Gamete Cryopreservation, Department of Medicine, University of Padova, 35128 Padova, Italy; gabriel.petre@rocketmail.com (G.C.P.); detoni.luca@gmail.com (L.D.T.); andrea.dinisio@gmail.com (A.D.N.); carlo.foresta@unipd.it (C.F.); 2Department of Medicine, Clinical Nutrition Unit University of Padova, 35128 Padova, Italy; francescofrancini@yahoo.it; 3Department of Women and Children’s Health, University of Padua, 35122 Padua, Italy; amerigovitagliano.md@gmail.com; 4Unit of Obstetrics and Gynecology, Madonna della Navicella Hospital, Chioggia, 30015 Venice, Italy

**Keywords:** fertility, ingredients, male reproduction, semen parameters, supplements

## Abstract

Dietary supplements (DS) represent a possible approach to improve sperm parameters and male fertility. A wide range of DS containing different nutrients is now available. Although many authors demonstrated benefits from some nutrients in the improvement of sperm parameters, their real effectiveness is still under debate. The aim of this study was to critically review the composition of DS using the Italian market as a sample. Active ingredients and their minimal effective daily dose (mED) on sperm parameters were identified through a literature search. Thereafter, we created a formula to classify the expected efficacy of each DS. Considering active ingredients, their concentration and the recommended daily dose, DS were scored into three classes of expected efficacy: higher, lower and none. Twenty-one DS were identified. Most of them had a large number of ingredients, frequently at doses below mED or with undemonstrated efficacy. Zinc was the most common ingredient of DS (70% of products), followed by selenium, arginine, coenzyme Q and folic acid. By applying our scoring system, 9.5% of DS fell in a higher class, 71.4% in a lower class and 19.1% in the class with no expected efficacy. DS marketed in Italy for male infertility frequently includes effective ingredients but also a large number of substances at insufficient doses or with no reported efficacy. Manufacturers and physicians should better consider the scientific evidence on effective ingredients and their doses before formulating and prescribing these products.

## 1. Introduction

Infertility is a pathological condition defined as the inability of a sexually active, non-contracepting couple to achieve pregnancy in one year [1]. Both male and female factors can lead to infertility. In particular, according to the causes, it has been reported that 29.3% is due to a male factor, 37.1% to a female factor, 17.6% to both male and female factors, with the remaining percentage considered as idiopathic [2].

It is estimated that around 10%–15% of all couples are affected by infertility, thus representing a global concern in most developed countries [3].

Among male infertility causes, many recent studies have emphasized the role of genital tract inflammation, incorrect lifestyles and malnutrition [4]. On this regard, weight excess and other conditions such as metabolic syndrome, alcohol abuse, cigarette smoking, exposure to environmental pollutants etc. have been strongly related to a decrease in sperm quality and fertility. A major driving hypothesis is that these conditions, by inducing an elevation of reactive oxygen species (ROS) and nitrogen species (RONS), are able to alter the balance of the redox status of both the steroidogenic cell population and the germ line cell populations, leading to the impairment of the hypothalamic–pituitary–testicular axis and the reduction of sperm quality [5].

A large number of recent studies have focused on the ability of many substances, generally termed as *nutraceuticals*, to improve the hormonal status and sperm parameters by different mechanisms [6]. Nutraceuticals are used as ingredients of dietary supplements (DS), widely marketed for the prevention or treatment of the most disparate pathological conditions. From a legislative point of view, the European Food Safety Agency (EFSA) defines that DS are not intended for the treatment or prevention of disease in humans, but only to support specific physiological function [7]. Currently, DS are widely prescribed to improve physiological aspects related to male fertility.

Many DS are available on the market with various formulations, containing both nutrients and botanicals at different doses. Despite many authors demonstrating positive effects of some ingredients on semen parameters and fertility outcomes [8], many others have also shown a lack of efficacy and even potentially harmful side effects [9]. In a recent position statement, the Italian Society of Andrology and Sexual Medicine (SIAMS) summarized the state of the art on each single ingredient currently used in the andrological field. In this paper, authors concluded that there was still limited scientific evidence on the possible role of any nutraceutical in andrology and the use of antioxidants could be suggested in patients with idiopathic infertility in the presence of documented abnormal sperm parameters only after a specific diagnostic workup. However, to date, no regulation or guidelines are available for the use of these products, generating confusion for both prescribers and patients [10]. Moreover, several factors make it difficult to empirically address the right ingredient for the right patient. In particular, it is difficult to identify the correct DS since each product contains different ingredients at different doses.

The purpose of this study was to critically evaluate the composition of DS employed in male infertility, using the Italian market as a sample.

## 2. Materials and Methods

In order to evaluate the potential efficacy of DS, a systematic literature review on substances used to improve sperm parameters was preliminarily performed. The literature search was conducted in MEDLINE, Scopus, EMBASE, and Cochrane Library registers until 31 March 2020. Only randomized clinical trials (RCTs) and systematic reviews or meta-analysis of RCTs were considered eligible. With the aim to rule out possible interactions between ingredients, only studies that used active substances alone or in combination with at most the other three ingredients were considered. The key terms used for the search were: fertility or male reproduction or semen parameters and supplements or ingredients. Figure 1 displays the flow diagram of the selection of eligible papers.

To establish the efficacy of each ingredient we considered only those having at least one RCT or systematic review or meta-analysis of RCTs, demonstrating a significant effect on any sperm parameters involved in male fertility. Significance was set at *p*-value < 0.05. When evaluating the findings of meta-analyses, we verified whether statistical methods incorporated substantial heterogeneity (Higgins I^2^ > 30%) into a random-effects model, as appropriate. Regarding the daily dose of each active ingredient with nutrient characteristics, we referred to the tolerable Upper intake Levels (UL) as reported in Dietary Reference Intake (DRI) [11].

Based on the results of available articles, we were able to identify, for each active ingredient, the minimal effective daily dose (mED) able to improve sperm parameters. To define mED we used the lowest effective dose reported in RCTs, systematic review or meta-analysis of RCTs. Therefore, we classified the ingredients contained in each supplement and suggested daily dose into three categories: reported efficacy with a dose achieving the mED (A), reported efficacy but with a dose below mED (B) and unreported data of efficacy (C). To classify DS, we created a formula taking into consideration the three classes of ingredients and their number:Score=(2A+B−C)2N×(A+B2)

In particular, the above formula was conceived based on the following sequential steps:(1)Each class of ingredients was given an arbitrary value: *A* = +2, *B* = +1 and *C* = −1;(2)These values were multiplied for the respective number of ingredients within each supplement (*A*, *B* and *C* respectively), obtaining a total score given by the sum of each category (2*A* + *B* − *C*);(3)As the number of ingredients highly differed between supplements, we standardized the above total score by dividing it for the maximum possible score for that supplement, by assuming that each ingredient was of class *A* (=*2N,* where *N* is the total number of ingredients in each supplement);(4)In order to correct this value for the number of ingredients of only categories *A* and *B*, the relative score was multiplied for the sum of high efficacy ingredients plus half (as a proxy of their lower efficacy) the number of moderate efficacy ingredients (*A + B/2*), finally obtaining a corrected score for each supplement.(5)Given the distribution of the scores resulted in three main clusters, we classified DS into three categories, resembling the efficacy of the ingredients: higher expected efficacy (corrected score ≥ 4), lower expected efficacy (4 < corrected score > 1) and no expected efficacy (corrected score ≤ 1).

We collected the names and formulations of the DS registered in Italy by referring to the register of the Italian Ministry of Health [12].

## 3. Results

The literature search on active ingredients allowed us to identify 41 studies (RCTs or meta-analyses) reporting their efficacy on sperm parameters (Figure 1). By this analysis we found that 18 of these ingredients had a reported efficacy. The complete list of ingredients with clinical evidence of efficacy, the respective references, evaluated sperm parameters and employed daily doses, are summarized in Table 1. In the right column, the mED of each ingredient is reported. In some studies, marked with an asterisk, the employed dose exceeded the reported UL. In particular, all the studies involving zinc evaluated the effect of this ingredient at a dose exceeding UL. For each active ingredient, the evidence of efficacy was supported by at least two RCTs or meta-analysis, excluding astaxanthin, D-aspartic acid and L-citrulline, which had only one reference.

Ingredients without clinical evidence in the improvement of sperm parameters (no RCT or meta-analyses) are listed in Table 2.

We found 21 DS marketed in Italy for male infertility. Their composition and the daily doses of their active ingredients are summarized in Table 3. Moreover, for each supplement, the scores of expected efficacy and the symbols summarizing the efficacy of their ingredients are reported.

A detailed analysis of this table raised the following considerations: (i) all supplements were mixtures of active ingredients; (ii) in each supplement the number of ingredients ranged from 2 up to 17, with a mean number higher than 7; (iii) 13 of 21 supplements contained at least one ingredient without reported efficacy; (iv) 19 supplements had ingredients below mED; (v) indeed, 1 supplement contained seven ingredients dosed below mED; (vi) 1 supplement contained only active ingredients satisfying mED; (vii) the product number 9 had a nutrient reaching UL (zinc 40 mg/day); (viii) zinc was the most used ingredient, followed by selenium, arginine, coenzyme Q, folic acid and carnitine. These substances were present in more than 50% of DS, whereas all the remaining ingredients were represented in 10% or less of products.

The distribution of DS into the three classes of efficacy is reported in Figure 2. Two DS out of 21 (9.5%) were included in the higher expected efficacy group. The majority of remaining products (71.4%) fell in the lower expected efficacy group, and four (19.1%) in the group with no efficacy.

## 4. Discussion

This critical review aimed to evaluate the formulation of supplements for male infertility using the Italian market as a sample. In general, there is still poor evidence in terms of large well-designed randomized and placebo-controlled trials availability, supporting the efficacy of nutraceutical products in the field of male reproductive health [54,55]. Nevertheless, these products are commonly administered to infertile patients [8,56]. Since a medical prescription is not necessary to purchase dietary supplements, subjects seeking fertility may have easy access to these products [10,57]. As a proof of concept, the Italian market of supplements generated 3.3 billion euros in 2019, with an increase of 4.3% compared to 2018 [58].

Whilst a rational use of supplements may be potentially beneficial for the improvement of sperm parameters, we need to stress that their uncontrolled use is potentially harmful for patients’ health due to direct toxic effects and interaction with drugs or nutrients [59]. In this respect, we were surprised to point out that all RCTs and meta-analyses on zinc for male infertility relied on doses always exceeding the UL. Over this background, in the near future it would be desirable to better define thoughtful criteria for each supplement in use.

Our analysis found that beside the gap of literature, the market of food supplements is still supported by poor scientific evidence. The majority of DS contained a huge number of ingredients, up to 17. The mixture of such a high number of ingredients may generate different issues, including a low concentration of each substance (i.e., necessitating of two or more administrations to reach the daily effective dose), a large volume of pills and a high risk of interactions. What is more, we found that some ingredients included in many DS had no scientific evidence of efficacy (i.e., astragalus, vitamin D3, taurine and riboflavin). The formulation of pills with a large number of ingredients, some of which cause uncertain benefits, denotes a gap of knowledge of potential biologic targets by manufacturers. Moreover, it has been reported that some plant extracts, present in many of these supplements, are likely to interact with drug metabolism [60,61]. This aspect raises further concerns on the safety of these products.

Very frequently, nutrients were present in DS at a dosage below mED. This situation was more common among products with a high number of ingredients. The administration of any active substance with a dose below mED appears as scientifically unjustified due to uncertainties in the therapeutic results. Differently, when the number of ingredients was small, the dose often satisfied mED. Another major aspect in the evaluation of supplements concerns safety. Some ingredients, particularly when administrated in high doses, are not free from risks when used as dietary supplements. For example, folates can mask the B12 deficiency favoring the progression of neurological damage [62]. The combination of these two vitamins could have a synergic effect in improving homocysteine metabolism hence the sperm quality. It should be noted that vitamin B12, when present, was rarely associated to folic acid [63,64]. Furthermore, zinc reduces the copper intestinal absorption interfering with its carrier [65]. With respect to this, we want to stress that one supplement on the market contained a dose of zinc reaching the UL.

On a positive note, our analysis revealed that some active ingredients with reported efficacy are frequently present in analyzed supplements. Previous studies demonstrated that some ingredients are particularly effective in specific patients’ conditions. Substances with antioxidant properties are indicated in inflammation of the male accessory glands, both related to microbial and non-microbial origin. Several studies performed in asthenozoospermic infertile patients, showed that the positive effect of selenium supplementation is dependent on the correct structure of the mitochondrial capsule [66,67]. Carnitine supplementation induced a significant increase in sperm motility in cases of asthenozoospermia with preserved mitochondrial function [68,69]. Due to the key role of zinc in the processes of DNA compaction, administration of this micronutrient was successful in improving sperm morphology and DNA integrity in patients with prostate abnormalities [70,71].

Based on active ingredients reaching mED we created a grading scale of supplements distinguishing three classes of expected efficacy. Three products were present in the higher class, some of which contained ineffective or underdosed ingredients. Most of the supplements fell in the lower group of expected efficacy. In this class, a large number of ineffective or underdosed products were also present. For an adequate evaluation of these classes, we considered the number of the effective ingredients as the most important criterion of efficacy. A relevant aspect was the use of ineffective or underdosed ingredients that should be absent or less than possible. Another parameter to evaluate a product was the presence of a lower number of ingredients.

We acknowledge the application of a non-validated statistical method to calculate scores for each DS may represent a point of weakness in this study. Very recently, a validated formula to score supplements was suggested by Kuchakulla et al. [72], based on the Budoff’s score, previously conceived by cardiologists to evaluate their procedures [73]. However, this scoring system when applied to DS, does not take into account the effective dose of ingredients, a crucial point in the evaluation of their efficacy. For example, using this approach, a DS containing ingredients at ineffective or toxic doses would be considered useful. As a point of strength, our scoring system relied on high quality evidence coming from RCTs or a systematic review and meta-analyses of RCTs, which represents a reliable approach to critically weighing the expected efficacy of dietary supplements. The same approach could be applied to evaluate products used in other clinical conditions.

In conclusion, this study showed that most DS marketed in Italy for male infertility contain ingredients with reported efficacy in the improvement of sperm parameters. Nevertheless, a non-negligible number of DS are mixtures of substances with uncertain or unreported benefits, whose administration may be unhelpful or even harmful for infertile patients. On that basis, we believe manufacturers should carefully scrutinize scientific evidence before delivering each supplements’ formulation. Accordingly, physicians should evaluate the composition of DS and the dose of each single constituent before considering their clinical use. Finally, the choice for DS should be tailored to the specific patient’s fertility problem.

## Figures and Tables

**Figure 1 nutrients-12-01472-f001:**
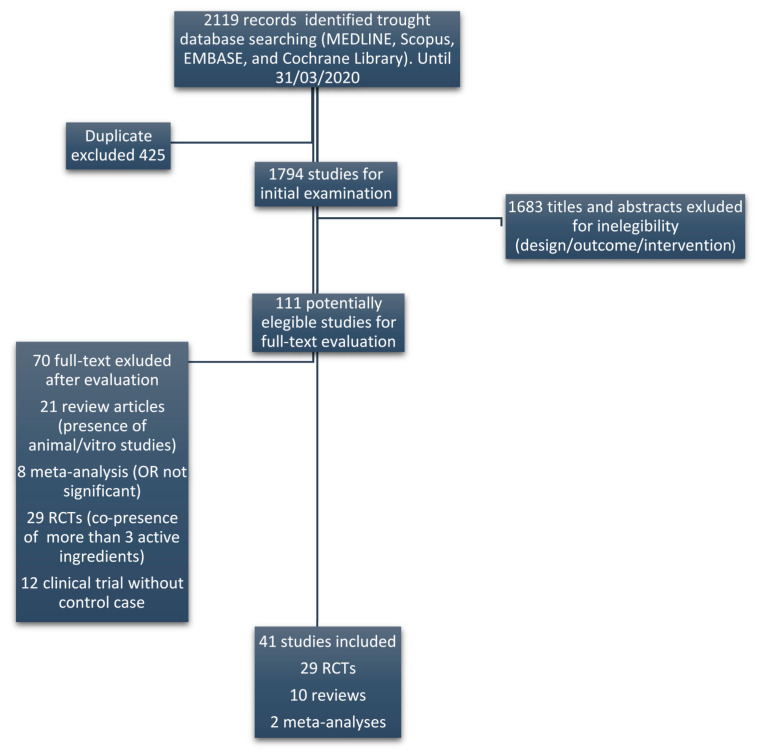
Flow diagram of the selection of eligible papers.

**Figure 2 nutrients-12-01472-f002:**
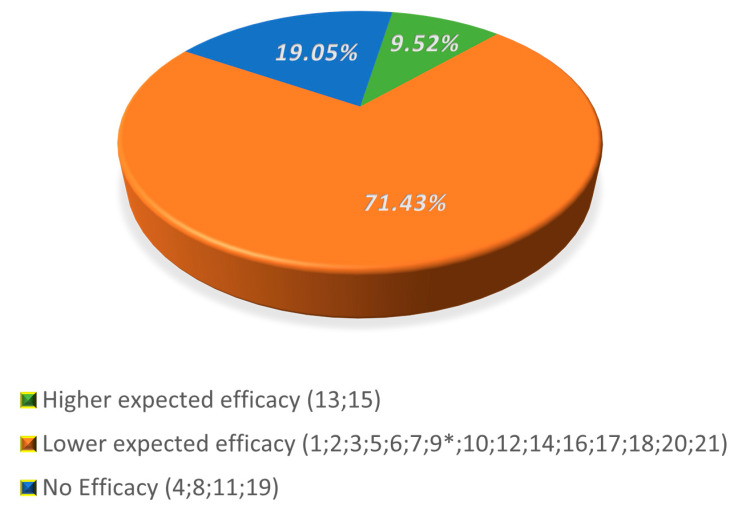
Distribution of supplements in classes of expected efficacy. * This supplement has a content of Zinc reaching the UL. Numbers refer to a specific supplement.

**Table 1 nutrients-12-01472-t001:** Active ingredients with evidence of efficacy, references, evaluated sperm parameters, employed daily doses and minimal effective dose (mED).

Active Ingredients	References	Evaluated Sperm Parameters	Employed Daily Dose	Minimal Effective Dose (mED)
Zinc	[13] * (Rev)	concentration	50 mg	50 mg
	[14] * (RCT)	concentration	66 mg	
	[15] * (RCT)	morphology	66 mg	
Selenium	[13] (Rev)	linear progression	50 µg	50 µg
	[16] (Met)	concentration	100 µg	
	[17] (RCT)	concentration/motility	200 µg	
Vitamin B12	[18] (Rev)	count	25 µg	25 µg
	[19] (RCT)	count	1500 µg	
	[20] (RCT)	count	6000 µg	
Folic Acid	[21] (RCT)	count/motility	400 µg	400 µg
	[22] (RCT)	volume/motility	500 µg	
	[23] (Met)	DNA damage	500 µg	
L-Arginine	[24] (RCT)	progressive motility	1.4 g	1.4 g
	[25] (RCT)	concentration/motility	1.4 g	
L-Citrulline	[26] (RCT)	volume/concentration motility/vitality	1.2 g	1.2 g
*α*-Lipoic Acid	[27] (RCT)	concentration/motility	600 mg	600 mg
	[28] (Rev)	motility/morphology	600 mg	
L-Carnitine (LC/LAC)	[29] (RCT)	motility	1 g	1 g
	[30] (RCT)	count/motility	2 g	
	[31] (RCT)	concentration/motility	2 g	
	[32] (Rev)	motility	3 g	
N-Acetyl Cysteine (NAC)	[33] (Rev)	motility/DNA damage	600 mg	600 mg
	[34] (RCT)	motility/DNA damage	600 mg	
	[17] (RCT)	motility/morphology	600 mg	
Coenzyme Q10	[35] (RCT)	motility	200 mg	200 mg
	[36] (RCT)	count/motility	200 mg	
	[37] (RCT)	concentration/morphology	300 mg	
Astaxanthin	[38] (RCT)	motility	16 mg	16 mg
D-Aspartic Acid (DAA)	[39] (RCT)	count/motility	2.7 g	2.7 g
Tribulus Terrestris DE	[40] (RCT)	count/motility	250 mg	250 mg
	[41] (RCT)	morphology/motility	500 mg	
	[42] (Rev)	count/morphology	6000 mg	
Myoinositol	[43] (Rev)	motility	2 g	2 g
	[44] (Rev)	concentration	4 g	
*α*-Tocopherol	[45] (RCT)	motility/DNA damage	20 mg	20 mg
	[32] (Rev)	motility/morphology	268.46 mg	
	[46] (RCT)	motility/lipid oxidation	300 mg	
	[47] * (RCT)	DNA damage	1000 mg	
Vitamin C	[48] (RCT)	concentration/motility DNA damage	0.5 g	0.5 g
	[49] (Rev)		1 g	
	[47] (RCT)		1 g	
EPA + DHA EPA + DHA	[50] (RCT)	concentration/motility	0.72 g + 0.48 g	DHA 0.48 g
	[51] (RCT)	DNA damage	0.14 g + 1 g	
Lycopene	[52] (RCT)	concentration/motility	4 mg	4 mg
	[53] (Rev)	count/morphology	8 mg	

* The employed dose exceeded/reach UL. LC: L-Carnitine; LAC: Acetyl L-Carnitine; EPA: Eicosapentaenoic acid; DE: Dry Extract; DHA: docosahexaenoic acid. Rev: Review; Met: Meta-analysis.

**Table 2 nutrients-12-01472-t002:** Ingredients without clinical evidence of efficacy.

Astragalus DE
Damiana DE
Nettle DE
Catuba DE
Ecklonia bicyclis DE
L-Taurine
Glutathione
Glucosamine
SOD
Vitamin D3
Vitamin B1
Riboflavin
Niacin
Vitamin B5
Vitamin B6
Biotin
Manganese

DE: Dry Extract; SOD: super oxide dismutase.

**Table 3 nutrients-12-01472-t003:** List of dietary supplements (**DS**) and relative composition.

**Active Ingredients**	**DS 1**		**DS 2**		**DS 3**		**DS 4**		**DS 5**		**DS 6**		**DS 7**	
	**S = 3.12**		**S = 2.08**		**S = 3.66**		**S = 0.16**		**S = 2.1**		**S = 2.25**		**S = 3.37**	
	**Daily Dose**	**EV**	**Daily Dose**	**EV**	**Daily Dose**	**EV**	**Daily Dose**	**EV**	**Daily Dose**	**EV**	**Daily Dose**	**EV**	**Daily Dose**	**EV**
**Zinc**	7.5 mg	**B**	10 mg	**B**	12.5 mg	**B**	1.5 mg	**B**			13 mg	**B**		
**Selenium**	60 µg	**A**			83 µg	**A**					30 µg	**B**	55 µg	**A**
**Vitamin B12**							33 mg	**A**						
**Folic Acid**	200 µg	**B**					400 µg	**A**			400 µg	**A**	200 µg	**B**
**L-Arginine**	100 mg	**B**			1000 mg	**B**			2500 mg	**A**	125 mg	**B**	30 mg	**B**
**L-Citrulline**														
***α*-Lipoic Acid**							50 mg	**B**						
**L-Carnitine**					1000 mg	**A**					200 mg	**B**	30 mg	**B**
**N-Acetyl Cysteine (NAC)**														
**Coenzyme Q10**	10 mg	**B**	200 mg	**A**	10 mg	**B**			200 mg	**A**	7.5 mg	**B**		
**Astaxanthin**	15 mg	**B**												
**D-Aspartic Acid (DAA)**			2660 mg	**A**										
**Tribulus terrestris DE**					800 mg	**A**								
**Myoinositol**													1000 mg	**A**
***α*-Tocopherol**	30 mg	**A**			12 mg	**B**			30 mg	**A**	36 mg	**A**	30 mg	**A**
**Vitamin C**	60 mg	**B**							180 mg	**B**				
**DHA**														
**Lycopene**					15 mg	**A**								
**Astragalus DE**					300 mg	**C**								
**Damiana DE**														
**Nettle DE**														
**Catuba DE**														
**Ecklonia bicyclis DE**														
**L-Taurine**									500 mg	**C**				
**Glutathione**							30 mg	**C**			40 mg	**C**		
**Glucosamine**														
**SOD**							154 UI	**C**						
**Vitamin D3**														
**Vitamin B1**														
**Vitamin B2**							25 mg	**C**						
**Vitamin B3**							36 mg	**C**						
**Vitamin B5**														
**Vitamin B6**							9.5 mg	**C**						
**Biotin**														
**Manganese**														
**Active Ingredients**	**DS 8**		**DS 9**		**DS 10**		**DS 11**		**DS 12**		**DS 13**		**DS 14**	
	**S = 1**		**S = 2.81**		**S = 2.66**		**S = 0.5**		**S = 1.14**		**S = 4.32**		**S = 2.5**	
	**Daily Dose**	**EV**	**Daily Dose**	**EV**	**Daily Dose**	**EV**	**Daily Dose**	**EV**	**Daily Dose**	**EV**	**Daily Dose**	**EV**	**Daily Dose**	**EV**
**Zinc**			40 mg	**B**	12.5 mg	**B**			10 mg	**B**			15 mg	**B**
**Selenium**			60 µg	**A**					55 µg	**A**	55 µg	**A**	83 µg	**A**
**Vitamin B12**									2.5 µg	**B**			5 µg	**B**
**Folic Acid**			800 µg	**A**					400 µg	**A**	200 µg	**B**		
**L-Arginine**	200 mg	**B**	250 mg	**B**	100 mg	**B**					30 mg	**B**		
**L-Citrulline**					800 mg	**B**								
***α*-Lipoic Acid**									300 mg	**B**			800 mg	**A**
**L-Carnitine**	200 mg	**B**	400 mg	**B**					500 mg	**B**	30 mg	**B**		
**N-Acetyl Cysteine (NAC)**									300 mg	**B**	600 mg	**A**		
**Coenzyme Q10**	15 mg	**B**	15 mg	**B**	100 mg	**B**			20 mg	**B**			200 mg	**A**
**Astaxanthin**														
**D-Aspartic Acid (DAA)**					80 mg	**B**								
**Tribulus terrestris DE**							300 mg	**A**						
**Myoinositol**							1000 mg	**A**	100 mg	**B**	1000 mg	**A**	1000 mg	**A**
***α*-Tocopherol**	30 mg	**A**	120 mg	**A**	30 mg	**A**					30 mg	**A**		
**Vitamin C**														
**DHA**														
**Lycopene**									4 mg	**A**				
**Astragalus DE**														
**Damiana DE**														
**Nettle DE**														
**Catuba DE**	50 mg	**C**												
**Ecklonia bicyclis DE**							200 mg	**C**						
**L-Taurine**														
**Glutathione**			80 mg	**C**										
**Glucosamine**							150 mg	**C**						
**SOD**														
**Vitamin D3**														
**Vitamin B1**									1.1 mg	**C**				
**Vitamin B2**									1.4 mg	**C**			2.8 mg	**C**
**Vitamin B3**									16 mg	**C**				
**Vitamin B5**									6 mg	**C**				
**Vitamin B6**									1.4 mg	**C**			2.8 mg	**C**
**Biotin**									100 µg	**C**				
**Manganese**									2 mg	**C**				
**Active Ingredients**	**DS 15**		**DS 16**		**DS 17**		**DS 18**		**DS 19**		**DS 20**		**DS 21**	
	**S = 4.33**		**S = 2.45**		**S = 2.06**		**S = 2.06**		**S = 1**		**S = 1.05**		**S = 2**	
	**Daily Dose**	**EV**	**Daily Dose**	**EV**	**Daily Dose**	**EV**	**Daily Dose**	**EV**	**Daily Dose**	**EV**	**Daily Dose**	**EV**	**Daily Dose**	**EV**
**Zinc**	15 mg	B	22.5 mg	**B**	10 mg	**B**	10 mg	**B**	6.5 mg	**B**	10 mg	**B**		
**Selenium**	50 µg	**A**			80 µg	**A**	50 µg	**A**			55 µg	**A**		
**Vitamin B12**	2.5 µg	**B**					1.5 µg	**B**						
**Folic Acid**	400 µg	**A**	300 µg	**B**	200 µg	**B**	200 µg	**B**					400 µg	**A**
**L-Arginine**			2500 mg	**A**			200 mg	**B**			30 mg	**B**		
**L-Citrulline**	3000 mg	**A**							200 mg	**B**				
***α*-Lipoic Acid**														
**L-Carnitine**	1000 mg	**A**	200 mg	**B**			400 mg	**B**			44,7 mg	**B**		
**N-Acetyl Cysteine (NAC)**														
**Coenzyme Q10**	200 mg	**A**					100 mg	**B**	90 mg	**B**				
**Astaxanthin**					16 mg	**A**	10 mg	**B**						
**D-Aspartic Acid (DAA)**									1000 mg	**B**				
**Tribulus terrestris DE**														
**Myoinositol**					50 mg	**B**					500 mg	**B**	4000 mg	**A**
***α*-Tocopherol**	40 mg	**A**	30 mg	**A**			12 mg	**B**						
**Vitamin C**	80 mg	**B**					100 mg	**B**			180 mg	**B**		
**DHA**					100 mg	**B**								
**Lycopene**	10 mg	**A**												
**Astragalus DE**														
**Damiana DE**	400 mg	**C**												
**Nettle DE**	300 mg	**C**												
**Catuba DE**														
**Ecklonia bicyclis DE**														
**L-Taurine**											300 mg	**B**		
**Glutathione**	40 mg	**C**					40 mg	**C**						
**Glucosamine**														
**SOD**					150 mg	**C**								
**Vitamin D3**							3.75 µg	**C**						
**Vitamin B1**														
**Vitamin B2**														
**Vitamin B3**														
**Vitamin B5**														
**Vitamin B6**														
**Biotin**														
**Manganese**														

S: score of supplement’s potential efficacy; EV: efficacy value of active ingredients; evidence of ingredients and dose efficacy: (A) reported efficacy and achievement of mED, (B) reported efficacy but below mED and (C) unreported efficacy. DE: Dry Extract; SOD: super oxide dismutase; DHA: docosahexaenoic acid.

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
