# Peer review of "Dietary Supplements for Male Infertility: A Critical Evaluation of Their Composition"

_nutrients, 2020, doi:10.3390/nu12051472_

Round 1
Reviewer 1 Report
This is an interesting review.
Some minor changes should be performed before it can be published.
- The formatting of the Abstract and some parts of the manuscript are wrong, please revise.
- Table 3 is much to Long and not very informative. I would suggest to either shorten it or put the complete table to supplementary Information.
- Figure 2 and especially the legend of Figure 2 is not informative. You should explain in the legend for what the numbers are Standing for. Maybe you should add a small table in Fig. 2 for the most promising DS as a Kind of summary.
Author Response
Thank you for allowing as to review this manuscript.
Dear Reviewer 1, thank you for allowing as to review this manuscript.
REVIEWER #1 Comment #1
The formatting of the Abstract and some parts of the manuscript are wrong, please revise.
Response: the abstract has been formatted and the manuscript completely revised.
REVIEWER #1 Comment #2
Table 3 is much to Long and not very informative. I would suggest to either shorten it or put the complete table to supplementary Information.
Response: table 3 has been improved and newly formatted. In our opinion, the new version is more informative and easier to read.
REVIEWER #1 Comment #3
Figure 2 and especially the legend of Figure 2 is not informative. You should explain in the legend for what the numbers are Standing for. Maybe you should add a small table in Fig. 2 for the most promising DS as a Kind of summary.
Response: the legend of figure 2 and footnotes have been improved in accordance, explaining that each number represents a DS. Information regarding most promising DS are now evident.

Reviewer 2 Report
The paper entitled "Food supplements in the treatment of male infertility: a critical review on their formulations and use" by Andrea Garolla et al. critically reviewed the composition of DS using the Italian market. The manuscript is interesting, but some points listed below should be considered before publication.
Whole text.
I really do not like using the word "proven" in scientific language. I know that you meant, but this word should be using highly carefully.
Similarly, "minimal FERTILITY effective dose". These doses have no relation to the reproductive outcome, but only semen parameters quality (unless research presents such an endpoint). Nevertheless, please describe a shortly mFED in the methods section.
The text of the manuscript was prepared carelessly (referring to editing the text).
Some small text changes suggestions:
Line 42: worse -> decrease in
Line 66: ad -> and
Line 61: the most part -> majority
Line 167: article -> critical review
Aim:
The aim is not clear and precise. It should be rephrased.
Materials and methods:
It should add information, does chosen supplements (mentioned lines 72-73) are described as that which are recommended especially to improve male fertility or semen quality.
The beginning of the methods is highly misleading. It is easy to focus only on the description of the systematic review. Also, even if you briefly describing systematic review it should be done lege artis. I do not expect the PRISMA tool form, but especially in this place, in the beginning, you should provide all important information. I am thinking about: 1) correct form of search key notation (presented search key does not present a systematic search strategy), 2) precise description of included research types, that was a meta-analysis of RCT ? and what type of reviews it was? (I know you mentioned it later but it is crucial to establish it at the beginning, one the same structure should be used throughout the article text).
When did you evaluate the DS effect on semen parameters on the basis of meta-analysis had you use only p-value? What about heterogeneity?
Moreover, I think it needs to be taken into account that semen quality alone is a poor predictor of fertility and male reproductive performance. It should be supported by clinical endpoints of reproductive outcomes, as live birth rates and clinical pregnancy. Did you consider that when you have chosen the research to include?
I am interested in which DS ingredients were categorized to which category (ABC). Could you provide this information as supplementary material?
Results:
Part of the results is put in footnotes of the "last" table 3 (lines 148-159).
Figure 1. and its description in results
A number of excluded (70) and included (42) studies not sum to 113 potentially eligible.
Does "12 no control case" refers to RCTs or other types of studies?
How many studies were RCT and how many meta-analyses?
Table 1.
Since RCT and meta-analyses or systematic reviews were used, I think you should present a type of study of in the table.
The title should be short and informative. All additional descriptions (abbreviations, asterisk) should be placed in table footnotes.
Table 3.
You should consider the horizontal position of table 3. The smaller font also should be used.
The title should be short and informative. All additional descriptions (abbreviations, asterisk, etc.) should be placed in table footnotes.
If the table is shared between pages you should provide a description for each of them (cont. Table 3)
In footnotes, you present part of the results.
I am not sure that a graphic representation (faces) is suitable. It would be better to write under or above the score that it is higher lower or no EE.
Figure 2.
Font of EE description should be smaller
To conclude, this critical review presents interesting insightful in DS effect on male semen parameters and indirectly on fertility. Its application potential is high and important for a different sector of health service. However, for the reasons mentioned above this paper requires reanalysis before considering its publication in the journal.
Thank you.
Author Response
Dear Reviewr 2, thank you for suggestions, allowing us to improve this manuscript.
Reviewer#2 Comment #1
I really do not like using the word “proven” in scientific language. I know that you meant, this word should be using highly carefully.
Response: Thank you for your comment. We now replaced the word “proven” with “reported” and other words such as “demonstrated”, “revealed”, “evidenced”, “showed”.
Reviewer#2 Comment #2
Similarly, “minimal fertility effective dose”. These doses have no relation to the reproductive outcome, but only semen parameters quality. Nevertheless, please describe a shortly mFED in the methods section.
Response: We acknowledge that the definition of “fertility” in relation to sperm parameters was potentially misleading for readers. All the text (including the title) was modified in line with your suggestion. Additionally, mFED was now changed in mED (minimum effective dose) and we better described it in the methods section, explaining that it is related to the improvement of sperm parameters.
Reviewer#2 Comment #3
The text of the manuscript was prepared carelessly (referring to editing the text)
Response: The entire text was now edited by an English native speaker and text changes suggestions were performed in accordance.
Reviewer#2 Comment #4
The aim is not clear and precise. It should be rephrased
Response: Thank you for your relevant observation. We now better clarified the aim of our study.
Reviewer#2 Comment #5
- It should add information, doses chosen supplements are described as that which are recommended especially to improve male fertility or semen quality. The beginning of the methods is highly misleading. You should provide the following information: 1) correct form of search key notation 2) precise description of included studies (meta-analysis of RCTs. 3)What about heterogeneity?
Response: Thank you for this comment. All the issues raised were now addressed in the revised version of the paper, better clarifying methods of search and evaluations of meta-analyses.
Reviewer#2 Comment #6
Semen quality alone is a poor predictor of fertility and male reproductive performance. It should be supported by clinical endpoints of reproductive outcomes.
Response: We acknowledge that semen quality is a poor predictor of fertility improvement. Unfortunately, we found a lack of evidence regarding the impact of DS on significant reproductive outcomes such as clinical pregnancy rates, ongoing pregnancy rates, live birth rates. Nevertheless, we hope you will agree that the demonstration of any improvement of semen parameters obtained by means of DS might represent a good scientific basis for postulating benefits on the reproductive outcomes. For this reason, we sincerely hope you will appreciate our efforts to shed some light on such a “grey area” of reproductive medicine.
Reviewer #2 Comment #7
I am interested in which DS ingredients were categorized to which category (ABC). Could you provide this information as supplementary material?
Response: this information is now provided in new table 3.
Reviewer #2 Comment #8
Results: Part of the results is put in footnotes of the "last" table 3 (lines 148-159).
Response: Sorry for the bad quality of previous PDF. In that file there was a problem during the PDF built with the formation of a wrong layout. In the next version we will carefully check the layout.
Reviewer #2 Comment #9
Figure 1. and its description in results:
A number of excluded (70) and included (42) studies not sum to 113 potentially eligible.
Response: Sorry for the typo error. It was corrected in accordance.
Does "12 no control case" refers to RCTs or other types of studies?
Response: 12 papers referred to studies with no control case.
How many studies were RCT and how many meta-analyses?
Response: in new figure 1 we reported the type of each study.
Reviewer #2 Comment #10
Table 1.
Since RCT and meta-analyses or systematic reviews were used, I think you should present a type of study of in the table.
Response: this information is provided in new table 1.
The title should be short and informative. All additional descriptions (abbreviations, asterisk) should be placed in table footnotes.
Response: the title has been changed in accordance. Abbreviations and asterisk have been placed in footnotes
Reviewer #2 Comment #11
Table 3. You should consider the horizontal position of table 3. The smaller font also should be used. The title should be short and informative. All additional descriptions (abbreviations, asterisk, etc.) should be placed in table footnotes. If the table is shared between pages you should provide a description for each of them (cont. Table 3) In footnotes, you present part of the results. I am not sure that a graphic representation (faces) is suitable. It would be better to write under or above the score that it is higher lower or no EE.
Response: Table 3 has been changed as suggested. Because your concerns regarding graphic representation (faces) we moved this information to figure 2.
Reviewer #2 Comment #12
Figure 2. Font of EE description should be smaller.
Response: Font of EE has been reduced
To conclude, this critical review presents interesting insightful in DS effect on male semen parameters and indirectly on fertility. Its application potential is high and important for a different sector of health service. However, for the reasons mentioned above this paper requires reanalysis before considering its publication in the journal.
Response: Many thanks for your cheering conclusions

Reviewer 3 Report
The work is interesting and has great practical significance.
However, the authors have to present the criteria of
the systematic review in a more precise way.
The authors in the introduction state that supplementation can
be effective only in patients with idiopathic infertility and
whether this has been included in the selection criteria
for work ?. Could the authors explain why they took only studies that used
active substances alone or in combination with at most other
three ingridients when it is known that ds always contain more? of them
Authors at work should consider the context of using
the term 'infertility'. It is difficult to talk
about the efficacy of DS in the treatment of male infertility when they were not analyzed in relation to achived pregnancies It is worth analyzing the diet, which will allow to determine
the possibilities of supplementation impact and whether the nutritional status was included in
the research and whether it could affect
the results of the research?
Author Response
Dear Reviewer 3, thank you for suggestions, allowing us to improve this manuscript.
REVIEWER #3 Comment #1
The work is interesting and has great practical significance. However, the authors have to present the criteria of the systematic review in a more precise way.
Response: Thank you for this comment. All the issues raised were now addressed in the revised version of the paper, better clarifying criteria of the systematic review, methods of search and evaluations of meta-analyses.
The authors in the introduction state that supplementation can be effective only in patients with idiopathic infertility and whether this has been included in the selection criteria for work?
Response: thank you for interesting observation. Most enrolled studies considered only idiopathic infertility, excluding other causes. However, in some papers the cause of infertility was not investigated/reported.
Could the authors explain why they took only studies that used active substances alone or in combination with at most other three ingredients when it is known that ds always contain more of them.
Response: We considered only studies that used active substances alone or in combination with at most other three ingredients in order to better understand efficacy of simple ingredients, minimizing interactions among active substances.
REVIEWER #3 Comment #2
Authors at work should consider the context of using the term 'infertility'. It is difficult to talk about the efficacy of DS in the treatment of male infertility when they were not analyzed in relation to achived pregnancies.
Response: We acknowledge that the definition of “fertility” in relation to sperm parameters was potentially misleading for readers. All the text (including the title) was modified in line with your suggestion. In accordance, mFED was now changed in mED (minimum effective dose) and we better described it in the methods section, explaining that it is related to the improvement of sperm parameters.
It is worth analyzing the diet, which will allow to determine the possibilities of supplementation impact and whether the nutritional status was included in the research and whether it could affect the results of the research?
Response: This is really an important item in the evaluation of the effects induced by DS. Unfortunately, this information was not provided by the reported literature.

Reviewer 4 Report
In this manuscript, the authors have presented their review on male infertility food supplement formulations in Italy market and scored them into 3 categories using non-validated method.
The presented data are sound. However, there are a few points that need to be addressed
- Keywords are not in alphabetical order.
- Typographical errors in Fig. 1: “trought”---through, at line number 94----formula tacking---taking.
- The table’s format is confusing. Should correct the format of tables. At line no. 145 a table has appeared without a title and again a same thing (Table without title) repeated at line no. 147.
- It would be better if authors would have applied a validated statistical method to calculate the scores.
- Relevant statistical analysis should be mentioned as footnote of tables and figures.
- Other citations are needed to strengthen the importance and relevance of the present results in the discussion section.
- Authors should check the conclusion part: Are the results really supporting the conclusion? The tested /selected parameters are sufficient for the conclusion?
- References should be cited by following journal style/format.
- Need to check for typographical errors, plagiarism, punctuation, and grammar throughout the manuscript.
Author Response
Dear Reviewr 4, thank you for suggestions, allowing us to improve this manuscript.
REVIEWER 4# Comment #1
Keywords are not in alphabetical order.
Response: Corrected in accordance
REVIEWER 4# Comment #2
Typographical errors in Fig. 1: “trought”---through, at line number 94----formula tacking---taking.
Response: the typographical errors has been changed as suggested.
REVIEWER 4# Comment #3
The table’s format is confusing. Should correct the format of tables. At line no. 145 a table has appeared without a title and again a same thing (Table without title) repeated at line no. 147.
Response: Sorry for the bad quality of previous PDF. In that file there was a problem during the PDF built with the formation of a wrong layout. In the next version we will carefully check the layout.
REVIEWER 4# Comment #4
It would be better if authors would have applied a validated statistical method to calculate the scores.
Response: we built a new formula to calculate the scores because we did not find any of appropriate at this aim. We recognize that it represents a limitations (as stated in the discussion section), however it seems a feasible tool to score DS.
REVIEWER 4# Comment #5
Relevant statistical analysis should be mentioned as footnote of tables and figures.
Response: the only used statistical analyses is related to the new formula.
REVIEWER 4# Comment #6
Other citations are needed to strengthen the importance and relevance of the present results in the discussion section.
Response: other citations have been added in the discussion section.
REVIEWER 4# Comment #7
Authors should check the conclusion part: Are the results really supporting the conclusion? The tested /selected parameters are sufficient for the conclusion?
Response: we carefully checked our conclusions. Even acknowledging the application of a new method to score the efficacy a DS, our opinion is that our conclusions are supported by results. If you have more specific observations, please write them to us.
REVIEWER 4# Comment #8
References should be cited by following journal style/format
Response: References have been typed following the journal style.
REVIEWER 4# Comment #9
Need to check for typographical errors, plagiarism, punctuation, and grammar throughout the manuscript.
Response: typo errors, punctuation and grammar were checked. No plagiarisms were found.

Round 2
Reviewer 2 Report
I would like to appreciate the efforts which authors have done in their work evaluated the effect of Italian marketed dietary supplements (included their doses) on semen parameters as a predictor of male fertility.
The changes implemented made are satisfactory. However, I still have some doubts about the final number of articles included in the review (112 minus 70 does not result in 41) show in Fig 1.
Thank you.
Author Response
REVIEWER #2 Comment #1
The changes implemented made are satisfactory. However, I still have some doubts about the final number of articles included in the review (112 minus 70 does not result in 41) show in Fig 1.
Response: Dear reviewer 2, many thanks you for comments and for identifying the typo error in figure 1 that we corrected in the new draft.

Reviewer 4 Report
In this manuscript, the authors have presented their review on male infertility food supplement formulations in Italy market and scored them into 3 categories using the non-validated method.
The authors had improved the manuscript but still, I feel there is a need for a statistical method to calculate the scores.
All other sections of the manuscript were improved.
Once again check for the references both in the text and reference sections are according to the journal format.
Need to check once again for typographical errors, plagiarism, punctuation, and grammar throughout the manuscript.
Author Response
Dear reviver 4,
Thank you for your consideration.
As stated in the first reply to reviewer’s comments, our method is based on a statistical approach even if not validated.
Searching for a validated formula, the only finding was a very recent paper by Kuchakulla et al. (February 2020), on the same topic using a formula derived from cardiologists of the American Heart Association to score procedures (Budoff et al. 2006).
In our revised manuscript we cited and discussed these references. Despite the paper by Kuchakulla et al. is very interesting and well designed, the Budoff’s formula doesn’t satisfy our needs because it does not consider the effective dose of ingredients representing a crucial point for DS efficacy.
Also, other concerns have been checked but we did not find further typographical punctuation, plagiarism and grammar errors through the manuscript. Anywhere, we are ready to further revise the manuscript in case you will indicate specific critical points.
